# Dual-Channel Pricing Decisions for Product Recycling in Green Supply Chain Operations: Considering the Impact of Consumer Loss Aversion

**DOI:** 10.3390/ijerph20031792

**Published:** 2023-01-18

**Authors:** Jiaying Xu, Qingfeng Meng, Yuqing Chen, Jia Zhao

**Affiliations:** School of Management, Jiangsu University, 301 Xuefu Road, Zhenjiang 212013, China

**Keywords:** green supply chain (GSC), product recycling, dual-channel pricing, loss aversion

## Abstract

With the vigorous rise of online third-party recycling platforms, dual-channel recycling has become the primary recycling mode in the reverse supply chain (RSC). However, as the main body of recycling, consumers have a significant impact on the recycling process, and their behavioral preferences are rarely considered in the pricing decision of the reverse recycling supply chain. Based on the dual-channel RSC, this paper considers the competition among channels. It introduces the loss aversion behavior preference of consumers to establish a dual-channel RSC composed of remanufacturers and online and offline recyclers. This study aims to analyze the impact of consumers’ loss aversion behavior on the recycling pricing and profit of each node in the green RSC and discuss the decision of recyclers under consumers’ loss aversion behavior. The results show that the deeper consumers’ aversion to the loss of recycling price, the lower the recycling price of dual-channel recyclers will be, which will be more conducive to the increase in the profit of online recyclers. However, the profit of remanufacturers will be reduced, and the total amount of recycling will decline. This paper considers the impact of consumer loss aversion behavior on dual-channel reverse supply chain pricing decisions based on prospect theory. It provides references for chain members to set recycling prices to increase people’s enthusiasm for recycling and the amount of recycled scrap, contributes to the cause of resource conservation and environmental protection, and improves the economic efficiency of recycling enterprises.

## 1. Introduction

With the rapid development of the social economy and the continuous expansion of consumer demand, the problems of resource shortage and environmental pollution are becoming increasingly severe. Green development has become a global consensus. In addition, the diversification of market demand, scientific and technological innovation, and other factors make product updating and iteration faster and faster, and its life cycle also shows a reverse trend of shortening. In particular, electronic products produce more and more waste products [1]. By 2021, the world will have generated about 57.4 million tons of e-waste, and the annual growth rate is 3.5% [2]. In China, only 26% of e-waste has been properly recycled. Moreover, the resource impact and potential health and environmental impact, two critical global issues, make e-waste a priority waste logistic [3], so the recovery and treatment of e-waste are extremely important and urgent. In order to alleviate environmental pressure, countries actively implement a circular economy to promote green and sustainable economic development. Sustainable development has become a problem that all sectors must face.

The traditional supply chain management is limited to fully utilizing resources within the supply chain. However, it needs to consider how to recycle the waste products of the whole supply chain and whether the supply process uses resources rationally, so the green supply chain has come into being. In order to promote the development of green supply chains, scholars, practitioners, and scientists from different fields have conducted research on how to promote the development of green supply chains. How to maximize the benefits of the supply chain members is a significant research problem to promote the development of a green supply chain, so many experts and scholars have researched the pricing decision problem of a reverse supply chain. For example, Wu et al. constructed a Stackelberg game model between recycling centers and third-party recyclers under centralized and decentralized decision-making to optimize recycling centers’ pricing and service decisions to obtain their optimal and maximum profits [4]. Shan et al. investigated the effect of cost-sharing strategies on the profits of remanufacturers, retailers, and third-party recyclers in a closed-loop supply chain [5]. Unlike the first two, Sui et al. consider the fair concern of recyclers and find that the fair concern factors will reduce the stable area of the system [6]. Li et al. take the heterogeneous behaviour of recyclers into account when studying the pricing decision of the reverse supply chain and find that the fair concern factors and the firm’s recycling behaviors, such as the speed of price adjustment, will affect the profitability of the system [7].

The above literature mainly studies recycling pricing decisions regarding recyclers’ behaviors without considering the influence of consumers’ behaviors. As one of the recycling subjects, consumers have an important influence on the behavioral preferences of reverse supply chain recycling and, therefore, gradually attract extensive attention from scholars. For example, Li and Wu et al. considered consumer preferences and service levels when designing a two-channel reverse supply chain multilevel network [8]. They found a positive relationship between consumer preferences for online channels and online recycling prices, profits, etc. Kang et al. proposed three pricing and service level decision models for online recycling centers and obtained consumer preferences through the Stackelberg game. The optimal pricing is obtained when consumer preferences change [9]. Unlike the first two, which considering consumer preferences, Zhu et al. integrate consumer bargaining power into the supply chain and establish a dual-channel closed-loop supply chain consisting of manufacturers, retailers, and online recycling platforms [10]. Zhao Jia et al. construct a two-channel reverse supply chain model under the dual influence of consumers’ bargaining power and recyclers’ loss aversion in the electronic recycling problem and make optimal decisions to improve the benefits of each member in the reverse supply chain [11]. Some scholars have also considered the impact of other consumer behaviors on the reverse supply chain. For example, Chen et al. combined consumer sustainability awareness and constructed a dual-channel reverse supply chain model for two cities. The study results showed that introducing online recycling channels and changes in consumer sustainability awareness would affect both cities’ pricing strategies and revenue [12]. Huang et al. considered the effect of strategic consumer behaviour on three remanufacturing scenarios. The study showed that more strategic consumers lead to a decrease in demand for new products and an increase in demand for remanufactured products [13]. Moreover, Das et al. considered consumer loss aversion behaviour in product return management. The above studies have adequately considered consumer behavioral factors in reverse supply chain studies [14]. However, little literature has examined the impact of consumer loss aversion behaviour on dual-channel recycling pricing decisions. In the dual-channel reverse recycling supply chain, consumers, as one of the main subjects of recycling, have loss aversion behaviors on recycling prices that can significantly affect the development of each node enterprise in the reverse supply chain.

When consumers face uncertain trading decisions, they often have expectations about prices and compare realized prices with reference points. Numerous studies in psychology and economics have shown that decision-makers are keener on minimizing losses than maximizing gains relative to reference points [15,16,17,18]. Prospect theory holds that for loss-averse decision-makers, a certain amount of losses will bring them more pain than the same amount of gains will please them [19] (Tversky and Kahneman, 1991). Aihuishou, China’s largest second-hand 3C electronic product recycling platform, is a typical example of online recycling. The actual recovery process is that after consumers choose the attributes of the products to be recycled, the platform gives a preliminary quotation, and the final recovery price will be fed back to consumers after the platform’s quality inspection is confirmed. In this recovery transaction process, consumers will face significant uncertainty because the price set by the enterprise is not necessarily the final transaction price. Secondly, consumers have expectations about the price of recycled goods. They compare the realized recycling price with the reference point. If the final recycling price is higher than expected, they will get satisfaction; otherwise, they will lose. Consumers’ loss aversion will significantly affect the recovery pricing decisions of enterprises. For example, Zhang et al. considered consumers’ loss aversion behaviour when studying the pre-sale pricing decision of the supply chain. They found that it would significantly impact the pricing decisions of supply chain members [20]. Liao et al. found that consumer loss aversion plays a vital role in critical decisions in the remanufacturing system. When faced with loss-averse consumers, manufacturers tend to raise product prices [21]. Wang et al. studied new product presale and returned strategies under strategic consumer loss aversion to providing pricing for retailers to presell new products and make optimal ordering decisions [22].

The above studies have significantly contributed to developing a green supply chain. Many scholars discuss the pricing decision of a reverse supply chain from the perspective of dual-channel recycling and fully prove that consumer behaviour has an important impact on the pricing decision of the supply chain. However, relevant researchers need to be more explicit about the mechanism of pricing decisions and the main body profit of dual-channel reverse supply chain under consumer loss aversion. Based on a dual-channel reverse supply chain, this paper constructs a utility function considering the loss aversion behaviour of consumers, discusses the amount of recycling in each channel, and analyzes the influence of changes in the degree of consumer loss aversion on the recycling pricing and profit of enterprises at each node of the reverse supply chain when the reference point of loss aversion is certain. Finally, it discusses how the profit of each node will change when consumers refer to different loss aversion reference points under the condition of a specific loss aversion coefficient. Compared with the existing studies, this paper makes some innovations and improvements in the following aspects:
(1)Compared with the existing literature, this paper studies the dual-channel recycle pricing from the perspective of competition based on the dual-channel reverse supply chain (RSC) and constructs a Stackelberg game model. It pays more attention to the impact of competition between actors on pricing and profits, enriches the management scenario of RSC pricing decisions, and provides management suggestions for enterprise development.(2)When studying the pricing decision of an RSC, most existing literature regards the decision-making subject of the supply chain as entirely rational. The research on consumers’ behavioral preferences has yet to be entirely carried out. Based on this, this paper considers the loss aversion behaviour of consumers in the pricing decision of dual-channel RSC, further enriching the research of RSC and making it closer to reality.

## 2. Problem Description and Model Construction

The dual-channel RSC studied in this paper comprises an online third-party recycling platform and an offline recycler, as shown in Figure 1. The remanufacturer is responsible for recycling, refurbishing, and remanufacturing and determines the recycling transfer price paid to the online third-party recycling platform and offline recycler. The online third-party recycling platform and the offline recycler recover waste from consumers at different recycling prices; there is a competitive relationship between the two.

In this paper, we study the Stackelberg game led by the remanufacturer under decentralized decision-making, where the remanufacturer is the leader, and the online third-party recycling platform and offline recyclers are the followers and competitors of each other. In this scenario, the remanufacturer is dominant in the RSC. It makes the decision first, while the online and offline recyclers make the corresponding decisions based on the manufacturer’s actions. The remanufacturer first decides the recycling transfer price $p_{t}$. After the remanufacturer completes the decision, the online and offline recyclers set their respective recycling prices $p_{a}$ and $p_{b}$.

### 2.1. Parameters and Symbol Definitions

The relevant parameters involved in this paper are shown in Table 1.

### 2.2. Related Assumptions

Considering the audience size of the online recycling channel and the fact that consumers need to deal with more significant uncertainty when recycling online, we hypothesized that consumers are more susceptible to loss aversion in the online recycling channel and developed a loss-averse consumer value assessment model. For simplicity, this study only considers a group of electronic products of the same type, brand, and degree of wear and tear, according to Chen and Feng et al. [12,23].

**Assumption 1:** The expected utility function is established. It is assumed that each consumer’s evaluation of waste products in the market is heterogeneous and uniformly distributed in a continuous range $\left[0,v\prime \right]$. The utility brought to the consumer by the recycling price is U(v)=p−v. When the utility is positive, consumers choose to recycle, and when the utility is negative, consumers do not participate in recycling. In addition, when there are online and offline recycling channels, consumers choose the more profitable recycling channel to recycle.

**Assumption 2:** Based on the fact that the recycling price of one unit of the used products in the offline recycling channel is $p_{b}$, then the consumer surplus recovered by the online recycling channel is $U_{b}=p_{b}-v-\omega$, where ω is the troublesome cost generated when offline recycling is inconvenient. As the waiting time in the recycling process is uncertain, consumers cannot participate in recycling anytime and anywhere, so this assumption is reasonable.

**Assumption 3:** Considering the relatively low operating cost of online platforms and online bidding mechanism, Feng et al. pointed out that the online recycling price of electronic products is generally higher than that of offline channels, that is, $p_{a}>p_{b}$ [24]. Based on the practice of Feng et al., this paper also assumes that $p_{a}>p_{b}$ is true. Similarly, according to Ma et al.’s hypothesis, the consumer surplus recovered by online recycling channels is $U_{a}=p_{a}-v-\mu$, where μ is the network and transportation costs that consumers need to pay when they participate in online recycling [25].

Then, the loss aversion valuation of the product introduced in the online recycling channel is shown in Figure 2.

As loss aversion is reference dependent, we set the consumer’s reference position to m. When the utility is greater than m, there is a gain, and when the utility is less than m, there is a loss. When loss-averse consumers want to avoid losses rather than gain, we give a lower weight to utilities above the reference point. Therefore, the cumulative utility distribution is bent at an angle γ, tanγ=k. The higher k is, the higher the degree of loss aversion. We obtain the utility function of consumers under loss aversion as
$$U_{a}=\left\{\begin{array}{cc} p_{a}-v-\mu & v<p_{a}-m-\mu \\ \frac{-1-k}{1-k}*v+\frac{1+k}{k-1}(\mu -p_{a})+\frac{2k}{k-1}*m & v\geq p_{a}-m-\mu \end{array}\right..$$

**Assumption 4:** Prospect theory suggests that the decision reference point determines the decision maker’s attitude toward risk. The heterogeneity of consumers determines that they will show different risk preference tendencies in the face of losses; therefore, this paper assumes that different consumers refer to different loss aversion reference points (LARPs) for the same recyclables.

### 2.3. Recycling Function of Dual Channel Reverse Supply Chain

We analyzed consumer utility based on different dual-channel recycling prices and defined the choice of consumer recycling channels as follows: choose the recycling channel that provides more efficient use to maximize consumer utility. If the utility of both recycling channels is negative, consumers do not participate in recycling. When the utilities of the two recycling channels are equal, consumers choose to recycle offline due to loss aversion to the price of the online recycling channel. The comparison of the utility of online and offline recycling channels is shown in Figure 3.
(1)As shown in Figure 3a, when $p_{a}-\mu <p_{b}-\omega$, $0<p_{a}-m-\mu <p_{a}-\mu$, the utility of offline recycling channels is always greater than online, regardless of whether consumers develop loss aversion; in the range of $\left(0,p_{b}-\omega \right)$, consumers choose offline channels to recycle; and in the range of $\left(p_{b}-\omega ,v\prime \right)$, consumers do not participate in recycling.(2)As shown in Figure 3b, when $p_{a}-\mu =p_{b}-\omega$, $0<p_{a}-m-\mu <p_{b}-\omega$, in the range of $\left(0,p_{a}-m-\mu \right)$, consumers tend to choose offline recycling as they are more likely to be loss averse to online recycling channels; in the range of $\left(p_{a}-m-\mu ,p_{b}-\omega \right)$, consumers choose offline recycling; and in the range of $\left(p_{b}-\omega ,v\prime \right)$, consumers do not participate in recycling.(3)As shown in Figure 3c, when $p_{a}-\mu >p_{b}-\omega$
and $0<p_{a}-m-\mu <p_{b}-\omega$, consumers choose online recycling in the range of $p_{a}$$\left(0,v_{0}\right)$, consumers choose online recycling in the range of $\left(v_{0},p_{b}-\omega \right)$, and consumers do not participate in recycling in the range of $\left(p_{b}-\omega ,v\prime \right)$.Here, $\frac{-1-k}{1-k}*v+\frac{1+k}{k-1}(\mu -p_{a})+\frac{2k}{k-1}*m=p_{b}-v-\omega$; that is, $v_{0}=\frac{-2km-\mu -k\mu +\omega -k\omega +p_{a}+kp_{a}-p_{b}+kp_{b}}{2k}$. As there are competitive behaviors between online and offline channels in the recycling process and to ensure the coexistence of dual channels, this paper assumes that the relevant parameters obey $p_{a}-\mu >p_{b}-\omega$, $0<p_{a}-m-\mu <p_{b}-\omega$. In the process of investigating the effect of consumer utility on the stability of the dual-channel recycling model, Ma et al. show that when offline consumer surplus is greater than online consumer surplus $p_{L}-\omega \theta -\theta >p_{H}-\mu -\theta$ but online consumer surplus is greater than the equilibrium point of online and offline consumer surplus $p_{H}-\mu >\frac{p_{L}+\mu -P_{H}}{\omega }$, offline channel sales will be equal to the equilibrium point of consumer surplus $v_{0}=\frac{p_{L}+\mu -P_{H}}{\omega }$, while online channel sales will be equal to online consumer surplus minus the equilibrium point $p_{H}-\mu -v_{0}$ [25]. In addition, Matsui et al. follow the same approach in their study of the optimal timing of price announcements for dual-channel RSC recovery [26].

In summary, referring to Ma and Matsui et al., assume that the recycling quantity of waste products is linearly distributed with the price, and we get the recycling volume of the online channel:$$q_{a}=v_{0}=\frac{-2km-\mu -k\mu +\omega -k\omega +p_{a}+kp_{a}-p_{b}+kp_{b}}{2k}.$$

The recycling volume in the offline channel is
$$q_{b}=p_{b}-\omega -v_{0}=\frac{2km+\mu +k\mu -\omega -k\omega -\left(1+k\right)p_{a}+\left(1+k\right)p_{b}}{2k}.$$

In the recycling process, online and offline recyclers deliver the recycled scrap to the remanufacturer, so the total amount of scrap recycled by the remanufacturer is equal to the sum of online and offline recycling: $q_{M}=q_{a}+q_{b}=p_{b}-\omega$.

## 3. Loss Aversion Dual Channel Recycling Model

In this paper, we study the Stackelberg game led by the remanufacturer under decentralized decision-making, where the remanufacturer is the leader, and the online third-party recycling platform and offline recyclers are the followers and competitors of each other. In this scenario, the remanufacturer is in the dominant position in the RSC and makes the first decision, and the online and offline recyclers then make the corresponding decisions based on the manufacturer’s actions. That is, the remanufacturer first decides the recycling transfer price of the used product $p_{t}$. After the re-manufacturer completes the decision, the online and offline recyclers set their own recycling prices $p_{a}$ and $p_{b}$. Then, according to the recycling volume of online and offline recycling channels obtained from the first part $q_{a}=\frac{-2km-\mu -k\mu +\omega -k\omega +p_{a}+kp_{a}-p_{b}+kp_{b}}{2k},q_{b}=\frac{2km+\mu +k\mu -\omega -k\omega -\left(1+k\right)p_{a}+\left(1+k\right)p_{b}}{2k}$, the profit function of the online and offline recyclers and remanufacturers can be obtained from the profit equation.

The loss aversion two-channel recycling model is
$$\left\{\begin{array}{c} \max\pi_{m}=(H-p_{t})*(q_{a}+q_{b})\\ s.t.\left\{\begin{array}{c} \max\pi_{a}=(p_{t}-p_{a}-c_{a})*q_{a}\\ \max\pi_{b}=(p_{t}-p_{b}-c_{b})*q_{b} \end{array}\right. \end{array}\right.$$

The online profit function is
(1)$$\pi_{a}=(p_{t}-p_{a}-c_{a})*q_{a}=(p_{t}-p_{a}-c_{a})*\frac{-2km-\mu -k\mu +\omega -k\omega +p_{a}+kp_{a}-p_{b}+kp_{b}}{2k}$$

The offline profit function is
(2)$$\pi_{b}=(p_{t}-p_{b}-c_{b})*q_{b}=(p_{t}-p_{b}-c_{b})*\frac{2km+\mu +k\mu -\omega -k\omega -\left(1+k\right)p_{a}+\left(1+k\right)p_{b}}{2k}$$

The remanufacturer’s profit function is
(3)$$\pi_{m}=(H-p_{t})*(q_{a}+q_{b})=(H-p_{t})*(p_{b}-\omega )$$

$\pi_{a},\pi_{b},\pi_{m}$ represent the profit of online and offline recyclers and remanufacturers, respectively; H represents the revenue after remanufacturing used products; and $C_{a}$ and $C_{b}$ are the recycling costs of online and offline recyclers, respectively.

### 3.1. Model Solving

In this paper, inverse regression is used to solve the optimal decision strategy of each party in the game. By calculating the first and second derivatives of Equations (1) and (2) concerning $p_{a},p_{b}$, we can obtain


$$\frac{\partial \pi_{a}}{\partial p_{a}}=\frac{2km+\mu +k\mu -\omega +k\omega -\left(1+k\right)c_{a}-2\left(1+k\right)p_{a}+p_{b}-kp_{b}+p_{t}+kp_{t}}{2k}\;\frac{\partial^{2}\pi_{a}}{\partial p_{a}^{2}}=-\frac{1+k}{k}\;\frac{\partial \pi_{b}}{\partial p_{b}}=\frac{-2km-\mu -k\mu +\omega +k\omega -\left(1+k\right)c_{b}+\left(1+k\right)p_{a}-2p_{b}-2kp_{b}+p_{t}+kp_{t}}{2k},\frac{\partial^{2}\pi_{b}}{\partial p_{b}^{2}}=-\frac{1+k}{k}$$


As 0<k<1, so $\frac{\partial^{2}\pi_{a}}{\partial p_{a}^{2}}<0,\frac{\partial^{2}\pi_{b}}{\partial p_{b}^{2}}<0$. Simultaneous $\frac{\partial \pi_{a}}{\partial p_{a}}=0,\frac{\partial \pi_{b}}{\partial p_{b}}=0$ can obtain:$$\left\{\begin{array}{c} \frac{2km+\mu +k\mu -\omega +k\omega -\left(1+k\right)c_{a}-2\left(1+k\right)p_{a}+p_{b}-kp_{b}+p_{t}+kp_{t}}{2k}=0\\ \frac{-2km-\mu -k\mu +\omega +k\omega -\left(1+k\right)c_{b}+\left(1+k\right)p_{a}-2p_{b}-2kp_{b}+p_{t}+kp_{t}}{2k}=0 \end{array}\right.$$

The solution is
(4)$$\left\{\begin{array}{c} p_{a}=-\frac{-2km-6k^{2}m-\mu -4k\mu -3k^{2}\mu +\omega -k^{2}\omega +2c_{a}+4kc_{a}+2k^{2}c_{a}+c_{b}-k^{2}c_{b}-3p_{t}-4kp_{t}-k^{2}p_{t}}{3+8k+5k^{2}}\\ p_{b}=-\frac{2km+\mu +k\mu -\omega -3k\omega +c_{a}+kc_{a}+2c_{b}+2kc_{b}-3p_{t}-3kp_{t}}{3+5k} \end{array}\right.$$

Take the first derivative with respect to $p_{t}$, and obtain the optimal solution:$$p_{t}^{*}=\frac{3H+3Hk+2km+\mu +k\mu +2\omega +2k\omega +\left(1+k\right)c_{a}+2\left(1+k\right)c_{b}}{6\left(1+k\right)}$$

Substituting $p_{t}^{*}$ into (4), the optimal solution of $p_{a},p_{b}$ is obtained as follows:$$p_{a}^{*}=\frac{9H+12Hk+3Hk^{2}+18km+38k^{2}m+9\mu +28k\mu +19k^{2}\mu +8k\omega +8k^{2}\omega -\left(9+20k+11k^{2}\right)c_{a}+8k\left(1+k\right)}{6\left(3+8k+5k^{2}\right)}$$
$$p_{b}^{*}=-\frac{-3H-3Hk+2km+\mu +k\mu -4\omega -8k\omega +\left(1+k\right)c_{a}+2\left(1+k\right)c_{b}}{6+10k}$$

### 3.2. Model Analysis

**Proposition 1:** *Offline channel recycling volume*$q_{b}$*is positively correlated with LARP*m*. Online channel recycling volume*$q_{a}$*and remanufacturer recycling volume*$q_{M}$*are negatively correlated with LARP*m.

**Proof:** $\frac{\partial q_{a}}{\partial m}=-\frac{3+7k}{9+15k}<0$; $\frac{\partial q_{b}}{\partial m}=\frac{3+4k}{9+15k}>0$;$\frac{\partial q_{M}}{\partial m}=-\frac{k}{3+5k}<0$. □

Proposition 1 shows that different consumers hold different levels of LARPs for the same salvage value. With the change in LARPs, the amount of offline channel recycling increases, while the amount of online channel recycling decreases, leading to a decrease in the total amount of recycling, i.e., the amount of recycling by the remanufacturer. For the same recycled product, at the same time, and in the same space, when consumers rely on a higher reference point, they are more likely to have loss aversion to the online recycling channel and turn to the offline recycling channel, which leads to a decrease in the online recycling volume and a corresponding increase in the offline recycling volume. In addition, compared with the actual recycling price, a higher reference point of loss aversion will lead to a stronger loss aversion of consumers, and the total recycling quantity, i.e., the remanufacturer’s recycling quantity, will be reduced.

**Proposition 2:** 
*The optimal recycling price*

$p_{a}$

*for the online channel and the optimal recycling transfer price*

$p_{t}$

*for the remanufacturer is positively correlated with the LARP*

m

*. The optimal recycling price*

$p_{b}$

*for the offline channel is negatively correlated with the LARP*

m

*. The optimal recycling price A for the offline channel is positively correlated with the LARP M.*


**Proof:** $\frac{\partial p_{a}}{\partial m}=\frac{18k+38k^{2}}{6\left(3+8k+5k^{2}\right)}>0$; $\frac{\partial p_{b}}{\partial m}=-\frac{2k}{6+10k}<0$; $\frac{\partial p_{t}}{\partial m}=\frac{k}{3\left(1+k\right)}>0$. □

Proposition 2 shows that a change in the LARP increases the optimal recycling price for the online channel and the optimal recycling transfer price for the remanufacturer and decreases the optimal recycling price for the offline channel. When consumers have higher LARPs for the same recyclables, online channel recyclers increase the channel recycling volume by raising the recycling price. Remanufacturers increase the total recycling volume by raising the recycling transfer price. At the same time, offline channel recyclers reduce the recycling price when consumers shift from online to offline recycling, resulting from the supply chain parties seeking to maximize benefits. Therefore, online and offline recyclers should pay close attention to the recovery market and consumers’ aversion to the loss of recovery price, stabilize the recovery price, actively improve the corporate image, and increase consumers’ willingness to recover to increase the amount of recovery and expand corporate profits.

## 4. Numerical Analysis

In order to study the effects of different loss aversion coefficients on the optimal recycling price, recycling volume, and profit of each party in a two-channel reverse supply chain, a numerical example of two-channel reverse supply chain pricing is given in this part of the paper, and Mathematica is used to analyze the sensitivity of the relevant parameters. By collecting the relevant literature, a broad range of parameters, such as the transportation cost of online recycling, trouble cost caused by the inconvenience of offline recycling, and loss aversion reference point, are obtained [24,26,27]. Moreover, take the recycling of iphone6 electronic devices on the love recycling platform as an example [12]. According to the above assumptions and the actual situation, the following assumptions are finally made for the parameters involved in the model:$$\mu =12,\omega =2,c_{a}=4,c_{b}=5,H=60,m=0.2$$

### 4.1. Effect of Loss Aversion Coefficient on Recycling Price

To ensure that each decision variable is non-negative, the loss aversion coefficients in this section are discussed in the range of 0.2<k<1.

The variation curves of recycling transfer prices, and online and offline recycling prices with different loss factors are shown in Figure 4.

Figure 4 shows that as consumer loss aversion deepens, recycling prices in online and offline channels decrease, while the recycling transfer prices of remanufacturers maintain a stable trend. This result is because online and offline recyclers are more susceptible to the influence of consumer loss aversion as they directly face consumers in the recycling process. When consumers’ loss aversion to online recycling prices deepens, online recycling platforms will lower recycling prices to reduce consumers’ losses to ensure their profits. At the same time, there are competitive behaviors between online and offline recycling channels in the recycling process. When the online recycling price decreases, offline recyclers will also adopt a lower recycling price to maximize profits.

Therefore, companies seek to maximize profits. When consumers unilaterally demand that losses are minimized, this leads to a loss of profit and efficiency for the company and the entire supply chain, resulting in a smaller business and a lower price for the final recycled product.

### 4.2. Effect of Loss Aversion Coefficient on Recycling Volume

As seen in Figure 5, when consumers are less averse to the loss of online recycling prices, online recycling increases more, while offline recycling decreases. Combined with Figure 4, in the range of 0.2<k<0.4, online recycling prices are much more significant than offline recycling prices, and the higher recycling prices are less different from consumers’ psychological expectations, which leads them to switch to online recycling channels that give them a minor loss. As consumers become more averse to the loss of recycling prices and the dual-channel competition further intensifies, the difference between online and offline recycling volumes decreases and balances out. As the difference in recycling prices between the two recycling channels is negligible in this case, there is little difference in consumer utility regardless of which recycling channel they choose.

In summary, with the aggravation of consumers’ loss aversion, the recovery of online channels will gradually increase, while offline channels will decrease progressively. However, due to the intensified competition between channels, the online and offline recovery prices will eventually become stable.

### 4.3. Impact of Loss Aversion Factor on Profit

From Figure 6, it can be seen that as consumers’ loss aversion to recycling prices deepens, the profits of the online recycling channel are on the rise, and the profits of the offline recycling channel remain stable. In contrast, the profits of remanufacturers have decreased. As consumers’ loss aversion to the online recycling channel deepens, the recycling transfer price of remanufacturers remains stable, but the recycling volume decreases, which indicates that the decrease in recycling volume is not conducive to the sustainable development of remanufacturers; although the profits of the online recycling channel are increasing, they are much smaller than those of the offline recycling channel in the range of 0.2<k<0.4. In combination with Figure 4, Figure 5 and Figure 6, it is found that with the deepening of consumers’ aversion to loss, the recycle price of offline and online recycle channels, as well as the profit and recycle volume of offline recyclers, are decreasing. Still, the recycling volume and profit of online recyclers are increasing. It shows that with the deepening of consumers’ loss aversion, online recyclers will gradually take advantage of the dual-channel RSC competition. But the overall supply chain recycling will decrease, which is not conducive to developing an RSC.

In summary, greater consumer loss aversion is conducive to lower recycling prices and increased profits and recycling volumes for online recyclers. However, it will lead to a decrease in the recycling volume and gain for offline recyclers and a reduction in the overall supply chain recycling volume.

### 4.4. Impact of Different LARPs on Profits

When consumers have different LARPs for the same recycled product, we analyzed the impact of the change in LARPs on the profits of the parties in the RSC in Table 2. Let the consumer’s loss aversion degree k=0.3, LARP $m_{1}=0.2,\;m_{2}=1$.

Table 2 shows that the profits of remanufacturers and online recycling channels decrease when consumers have a higher LARP for the same recyclables, while offline recycling channels increase. When the consumer-referenced LARP changes from 0.2 to 1, the optimal recycling price and recycling volume of each node firm in the RSC do not change significantly, and the changing trend of their respective profits is consistent with the changing trend of recycling volume. It can be seen that the change in consumer LARP is more favorable to an increase in profit when the enterprises increase the recycling volume.

## 5. Conclusions

Based on the background of socially sustainable development and the rapid development of Internet technology, online third-party recycling platforms are booming, and dual-channel recycling has become the primary recycling mode in the RSC. However, there is still the problem that consumers will have loss aversion when facing uncertain recycling prices. To solve this problem, this paper introduces the loss aversion coefficient and LARPs and constructs a loss aversion consumer model of dual-channel RSC based on channel competition. The research finds that:
(1)Online and offline recycling channels will lower the recycling price to optimize profits when customers use the same LARP for the same recycled goods. However, the overall volume of recycling will fall. As consumers’ loss aversion deepens, the recycling prices of dual-channel recyclers show a decreasing trend of change, while the recycling transfer prices of remanufacturers maintain a stable trend. Competitive behaviour exists between online and offline recycling channels in the recycling process. When the online recycling price decreases, offline recyclers will also adopt a lower recycling price to maximize profits. Therefore, recycling companies in a competitive market environment can improve their service level to enhance their strength as a “price war” is not desirable.(2)When consumers refer to different LARPs for the same recycled product, although the recycling volume, optimal recycling price, and profit of each node enterprise have changed, the range of change is small. The deepening of consumers’ loss aversion will reduce recycling volume by recyclers and result in a reduction in the total recycling volume as well. As a result, companies should actively improve their image and expand their credibility while lowering recycling prices, such as using advertising and policies to guide consumers to increase their willingness to recycle and their awareness of sustainability and promote the growth of recycling volume.(3)Regardless of the loss aversion coefficient change or LARP, each node enterprise’s profit change trend in the RSC is basically similar to the changing trend in recycling. Therefore, in the recycling process of waste products, online and offline recycling channels should reduce the degree of competition between channels, stabilize their recycling prices, strive to improve the amount of recycling, and effectively increase the revenue of each channel in the RSC. The deeper the loss aversion of consumers to recycling prices, the more favorable the profit increase of online recyclers, but the profit increase of offline recyclers is less, and the profit of remanufacturers is reduced. Therefore, when recycling, enterprises should pay attention to consumers’ loss aversion to recycling prices by understanding the recycling market situation, stabilizing the recycling price, and actively improving the corporate image to increase consumers’ willingness to recycle in order to increase the volume of corporate recycling and expand corporate profits.

The hypotheses of this study still need further research. First, this paper supposes that the remanufacturer can take all recyclables from recyclers. If the remanufacturer’s demand is uncertain, what impact will consumers’ loss aversion have on it? Secondly, different types of recycled goods (such as basic waste and second-hand luxury goods) have different residual values. How will consumers’ loss aversion change when recycling, and how will the profits of each node in the RSC change? Therefore, the future study can comprehensively consider the influence of consumers’ loss aversion on the pricing decisions of each node when the remanufacturer does not fully accept the recyclers’ recyclables and faces different types of recyclables.

## Figures and Tables

**Figure 1 ijerph-20-01792-f001:**
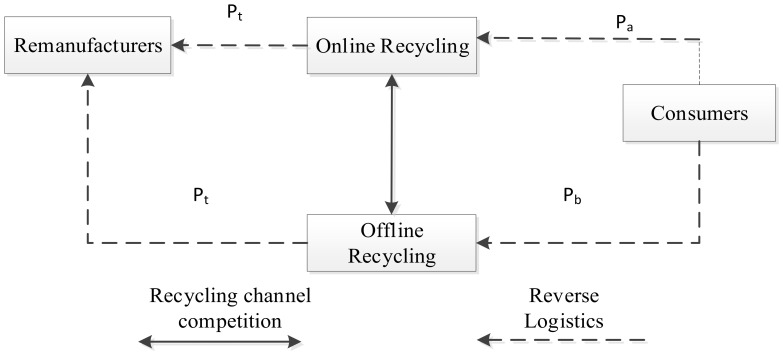
Dual-channel reverse supply chain model.

**Figure 2 ijerph-20-01792-f002:**
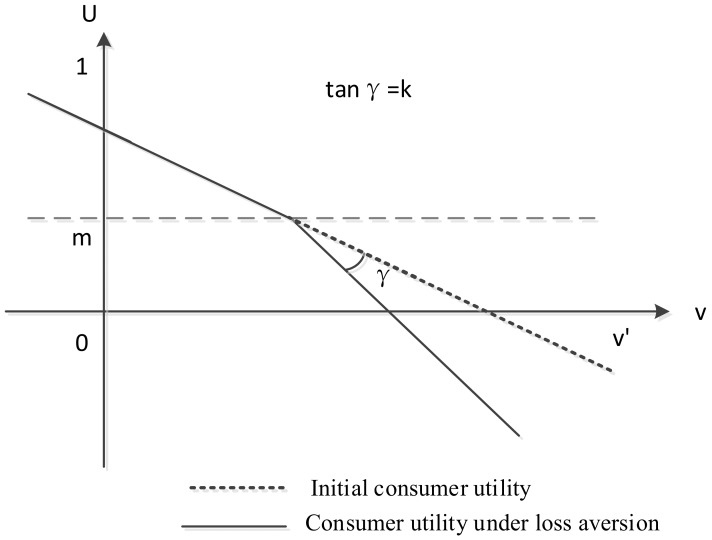
Consumer utility based on loss aversion reference point m.

**Figure 3 ijerph-20-01792-f003:**
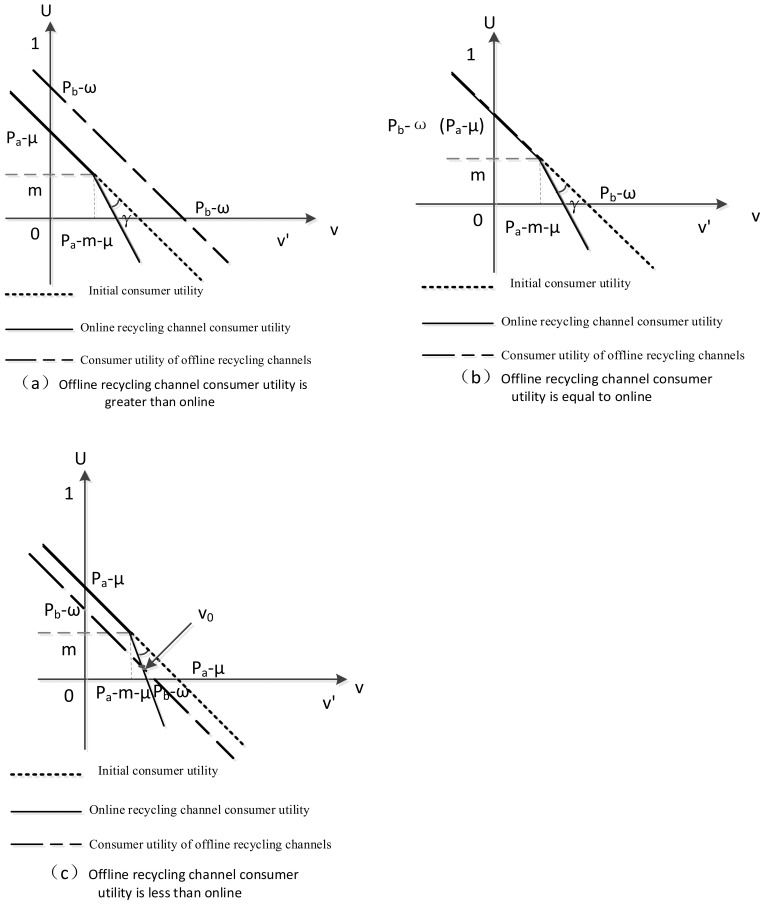
Consumer utility of dual-channel recycling under different scenarios.

**Figure 4 ijerph-20-01792-f004:**
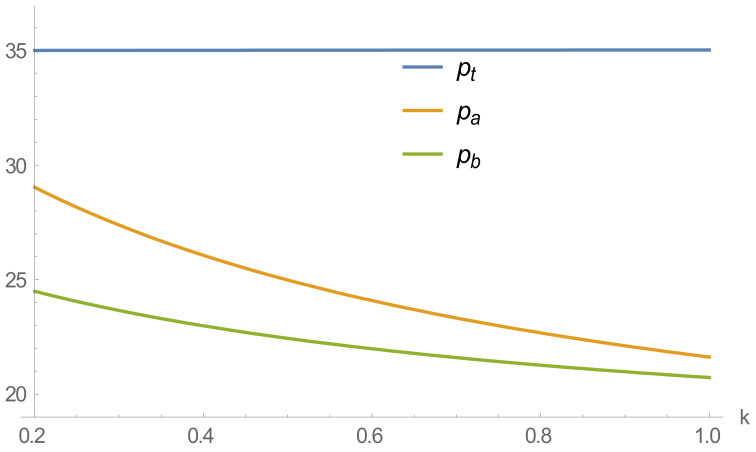
Effect of loss aversion coefficient on recycling price.

**Figure 5 ijerph-20-01792-f005:**
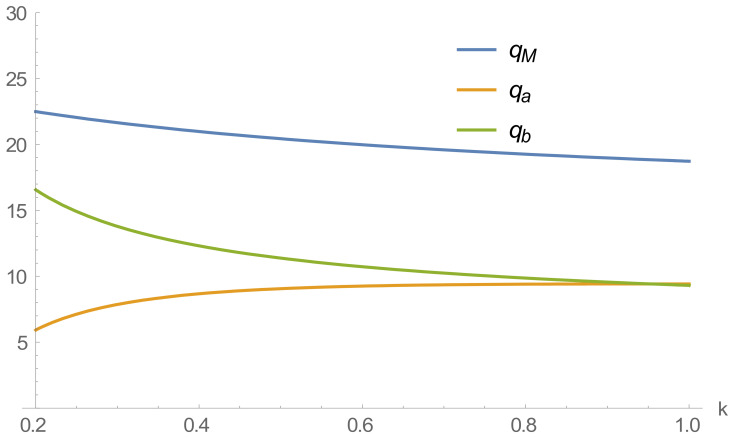
Effect of loss aversion coefficient on the amount of recycling.

**Figure 6 ijerph-20-01792-f006:**
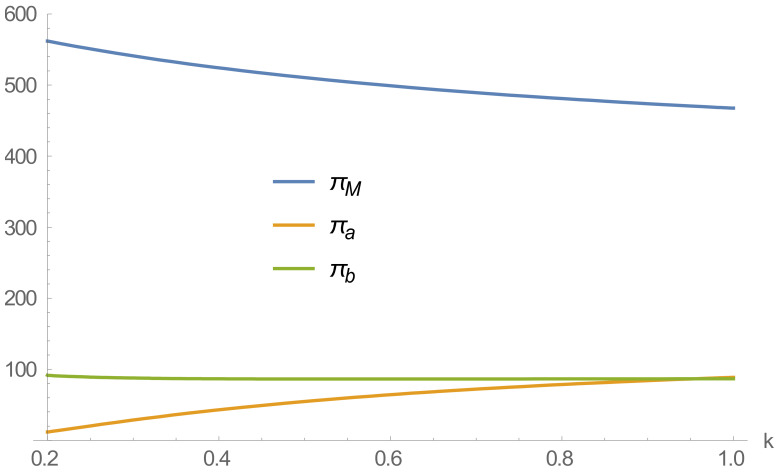
Effect of loss aversion coefficient on profit.

**Table 1 ijerph-20-01792-t001:** Summary of relevant symbols.

Symbols	Description
v	Consumer perceived value of used products. $v\in \left[0,v\prime \right]$
μ	Unit shipping costs paid by consumers participating in online recycling channels
ω	The cost of hassle caused by the inconvenience of consumer participation in offline recycling
m	Loss aversion reference point
k	Loss aversion. The larger the *k*, the higher the loss aversion. $k\in \left(0,1\right]$
H	Revenue from remanufacturing of unit waste products
$c_{a}$	Unit online recycling channel cost
$c_{b}$	Unit offline recycling channel cost
$q_{a}$	Online recycling demand
$q_{b}$	Offline recycling demand
$p_{t}$	Recycling transfer price
$p_{a}$	Online recycling prices
$p_{b}$	Offline recycling prices
$\pi_{a}$	Benefits for online third-party recyclers
$\pi_{b}$	Benefits for offline recyclers
$\pi_{M}$	Remanufacturers’ earnings

**Table 2 ijerph-20-01792-t002:** Numerical analysis of different loss aversion reference points.

	$m_{1}=0.2$			$m_{2}=1$		
	Remanufacturers	Online Channels	Offline Channels	Remanufacturers	Online Channels	Offline Channels
Recycling volume	21.7	7.9	13.8	21.6	7.6	14
The best recycling price	35	27.4	23.7	35.1	27.6	23.6
Profits	541	28.6	87.7	538	26.4	90.9

## Data Availability

Not applicable.

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
