# Peer review of "Dual-Channel Pricing Decisions for Product Recycling in Green Supply Chain Operations: Considering the Impact of Consumer Loss Aversion"

_ijerph, 2023, doi:10.3390/ijerph20031792_

Round 1
Reviewer 1 Report
This manuscript discussed a dual channel recycling model with the consideration of loss aversion. The topic is interesting; however, I have the following concerns.
1. The model shown in the manuscript needs to explain which products are suitable to it. Its validation is needed to be discussed.
2. Please explain how to derive the equation of Ua shown in page 5.
3. Figure 3 is not clear, such as where v0 is.
4. It is needed to explain why volume (qa) = value(v0), they have different units.
5. The meaning of profit equations can be guessed, but they need to be described.
6. Parameter values assigned as shown in Section 4 need to be discussed where they are from and their meanings.
7. The range of k needs to be discussed in terms of meaning.
Reviewer 2 Report
This manuscript constructs a dual-channel reverse supply chain model with consumers' loss aversion, and gets the optimal prices of remanufacturers and online and offline recyclers, which has a certain theoretical value.
However, the novelty of this manuscript is only reflected in the addition of a loss aversion parameter, and only price-decisions are mentioned for all the decision-makers.
In a word, the manuscript is not innovative enough to publish on the IJERPH.
Reviewer 3 Report
Comments on ‘Dual-Channel Pricing Decisions for Product Recycling in Green Supply Chain Operations: Considering the Impact of Consumer Loss Aversion’
In this paper the authors investigate the dual-channel reverse supply chain with one offline recycler and one online recycler, then consider the impacts of the remanufacturer's pricing decision and consumer's loss aversion on supply chain performance. The analysis focuses on the loss aversion coefficient and reference point, and discusses the equilibrium profits through solving a Stackelberg game model. Although the paper has done some work, there are several problems. I have the following criticisms:
1. The authors believe that consumers face higher uncertainty through online recycling channel and assume only the online consumers are loss-averse. However, it is not convincing. All the consumers are loss-averse when they sell possessions to the third-party firm no matter online or offline.
2. The authors assume the online recycling price is higher than the offline recycling price. However, in practice, most offline channels have higher searching-cost and transportation cost, which makes consumers are more willing to choose online recycler even if the recycling price is lower than the offline channel.
3. The Abstract is too weak to indicate the novelty of this research. First, the abstract should be simplified and include: one sentence clearly saying what is the current knowledge gap, one sentence explaining why this problem is worth to studied, one or two sentences discussing the results and drawing broader and more general conclusions.
4. The Introduction and Literature Review are insufficient to support the novelty. The introduction lacks the discussion of the importance and necessity of this topic. It is necessary to give more detailed explanations on what this research adds and summarizing the main findings clearly.
5. What is the innovation of this decision model? Is there any unexpected result in the research? All the conclusions and analyses are intuitive.
6. This paper greatly lacks the practical cases and applications, especially in Section 4, Numerical analysis. Moreover, the illustrations should be further discussed by highlighting that the manuscript contains sufficient contributions to the industrial practice and government regulation. The findings should be explained more.
7. The authors should try to present some punchline insight in conclusions and managerial suggestions in Section 3, 4 and 5. In other words, the authors should be presented no-included mathematical phrases that can help the readers who use the results, directly, as a managerial tool.
8. Attention had to be paid to the clarity of expressions and readability, such as sentence structure, grammars, acronyms and layouts. i.e. The sentence is incomplete in Page 3, Line 130.
Round 2
Reviewer 1 Report
The authors have answered all my concerns.
Reviewer 3 Report
NA
